# Roosters do not warn the bird in the mirror: The cognitive ecology of mirror self-recognition

Sonja Hillemacher[1]*, Sebastian Ocklenburg[2,3,4], Onur Güntürkün[2,5], Inga Tiemann[1]

1 Institute of Agricultural Engineering, University of Bonn, Bonn, Germany, 2 Institute of Cognitive Neuroscience, University of Bochum, Bochum, Germany, 3 Department of Psychology, MSH Medical School Hamburg, Hamburg, Germany, 4 ICAN Institute for Cognitive and Affective Neuroscience, MSH Medical School Hamburg, Hamburg, Germany, 5 Research Center One Health Ruhr, Research Alliance Ruhr, Ruhr University Bochum, Bochum, Germany

ʘ These authors contributed equally to this work.
* sonja.hillemacher@uni-bonn.de

**Data Availability Statement:** All relevant data are within the paper and its Supporting Information files.

**Funding:** Supported by the German Research Council (DFG) through Gu 227/16-1 (OG) and by

## Abstract

Touching a mark on the own body when seeing this mark in a mirror is regarded as a correlate of self-awareness and seems confined to great apes and a few further species. However, this paradigm often produces false-negative results and possibly dichotomizes a gradual evolutionary transition of self-recognition. We hypothesized that this ability is more widespread if ecologically tested and developed such a procedure for a most unlikely candidate: chickens (*Gallus gallus domesticus*). Roosters warn conspecifics when seeing an aerial predator, but not when alone. Exploiting this natural behavior, we tested individual roosters alone, with another male, or with a mirror while a hawk's silhouette flew above them. Roosters mainly emitted alarm calls in the presence of another individual but not when alone or seeing themselves in the mirror. In contrast, our birds failed the classic mirror test. Thus, chickens possibly recognize their reflection as their own, strikingly showing how much cognition is ecologically embedded.

## Introduction

Mirror self-recognition (MSR) is often considered a signature of self-awareness [1] and thus constitutes a unique experimental window into the animal mind. MSR studies make it likely that this ability is not necessarily an exclusively human feature, but in principle seems also to be present in some great apes like chimpanzees, bonobos, and orangutans [2, 3], bottlenose dolphins [4, 5], Asian elephants [6], Eurasian magpies [7] (but see [8]), Indian house crows [9], and cleaning wrasses [10–12]. In a graded form, which implies rather intermediate levels of mirror-understanding, this ability seems also to be present in macaques [13, 14], capuchin and spider monkeys [15, 16], nutcrackers [17], zebra finches [18], and pigeons [19].

Self-directed behavior occurring only and spontaneously in the mark-and-mirror-condition is assumed to indicate self-recognition. When first exposed to a mirror, most animals

ERC-2020-ADG, AVIAN MIND, LS5, GA No. 101021354 (OG). The funders had no role in study design, data collection and analysis, decision to publish, or preparation of the manuscript.

show social responses towards their mirror image like acting aggressively towards a conspecific [3]. In chimpanzees, social responses often decline with increasing time of mirror exposure, while contingent and spontaneous self-related behaviors increase in parallel [20]. This transition from social to self-directed behavior is an important component of MSR. Subsequently, a mark (only visible in a mirror) is applied to the animal's face or body. Behaviors directed to this mark are then interpreted as mark- and thus self-directed behavior and hence as a final proof of MSR. Control conditions include invisible sham marks matching the methodology of the application.

The last decade witnessed increasing controversies about the mark test. First, this test might indicate the ability of non-human animals to recognize the contingency of their own actions and their reflected behavior, although this might not necessarily implicate self-awareness [10, 17, 21]. Second, in all mark test studies with non-human animals, only a subset of individuals passes the test [6, 7, 22] while initially successful individuals don't necessarily pass during test repetitions [6]. Thus, either only a fraction of individuals of a species possess self-awareness, or the MSR-procedure produces high rates of false-negative results [21, 23, 24]. This is also found among human participants. Although the vast majority of children from industrialized countries pass the mark test when being 18 months or older, only very few children from the rural expanses of Kenya do so, pointing to important cultural factors that affect test outcomes [25]. Macaques, that generally do not pass the mark test [3], display positive test outcomes when trained beforehand with an irritant laser that can be seen in a mirror [13]. Notably, after having passed the mark test, these macaques make spontaneous use of mirrors to inspect parts of their body that they normally could not observe directly [13]. Thus, this simple training enables these animals to show spontaneous behavior akin to chimpanzees [3]. Capuchins and spider monkeys [15, 16] and pigeons [19] fail the mark test and do not show social interactions with their mirror image. Instead, they seem to see it as a strange conspecific rather than a stranger. These results show that the mark test has serious boundaries. If many chimpanzees don't pass the test [22], if culture affects the success of human children [25], and if simple training in macaques or changing the size of the mirror in chimpanzees can result in positive result patterns [13, 14, 26], it is likely that test outcomes are affected by a broad range of methodological and motivational factors that will increase the possibility of false negative results [23, 24, 27].

In addition, the mark test only produces binary fail-or-pass outcomes and doesn't capture the evolutionary much more likely graded result pattern observed by other means, which ascribes different levels of mirror-understanding to the different species according to their evolutionary, social and cognitive context [11, 15–17, 19, 28–30]. These limitations obviously challenge attempts to conceive evolutionary interpretations of self-recognition by using the classic mark test.

Therefore, we propose that MSR experiments should be embedded into the context of a species' ecological behavior [11, 31]. To this end, we used the alarm calling behavior and the corresponding audience effect as a natural behavioral pattern of chicken [32]. Typically, roosters react to the presence of a predator with an alarm call [33], depending on the predator as well as on the audience: different alarm calls for aerial and terrestrial predators are used [34] and roosters emit alarm calls most likely when they can warn an audience of females that could be mated or genetically related males [32, 35]. When they are alone or if there is a rivaling, non-related conspecific rooster, they will keep silent and thus reduce their own risk of being preyed [33].

With over 19 billion individuals worldwide used for meat and egg production each year, chickens are the most widely used farm animal. Despite their worldwide presence and use, only a few studies addressed their cognitive capacities [36].

In this study we applied two different testing methods of MSR on adult, male chickens. As fearfulness and predator-responses are known to be modified by domestication [37] and thus could be pronounced differently in chicken with different genetic backgrounds, we tested roosters of six different genotypes, ranging from hybrid lines to indigenous breeds. For the mirror-audience test, the first test, we utilized the natural behavior of roosters which emit an alarm call in the presence of an aerial predator when accompanied by a conspecific [32]. Roosters were kept in a compartment adjacent to another compartment of equal size and we compared the occurrence of alarm calls under different conditions: (A) no conspecific, (B) in front of a mirror that divided the two compartments, (C) in the presence of a conspecific male that could be seen in the adjacent compartment through an acrylic glass partition and (D) with a conspecific male hidden behind the mirror in the adjacent compartment as a control condition for olfactory and acoustical cues. This ecological testing method matches chickens' natural behavior and biological abilities and might be more appropriate to this species than the classic mark test. In the second test, we run the well-known, classic mark test with a subset of the individuals following the exact protocol of Prior et al. [7], who applied this test to magpies. Furthermore, we considered the critics of Clary and Kelly [17] on sticker markings by using powder markings to minimize tactile cues. The intention of this study was twofold: First, we aimed to see if male chickens perceive their reflection in the mirror as a conspecific or if they are capable of MSR. Second, by comparing the outcome of both procedures we wanted to test our hypothesis that embedding mirror self-recognition into the ecological framework of the animals can release situated cognitive abilities that are otherwise invisible.

## Materials and methods

### Experimental model and subject details

In the mirror-audience test, we conducted two identical experiments several years apart. First, we tested 50 roosters of four different breeds: 20 Rhinelander (RL), 10 Breda (BR), 10 Bergische Long Crower (BLC) and 10 roosters of Lohmann Brown breeders ($LB_b$). These animals have been raised and kept at the Poultry Research Centre, Rhein-Kreis Neuss or purchased as full-grown from local breeders several months before the experiments. They were under veterinary supervision. Each breed was kept under conventional free-range conditions with a barn (7.7 $m^2$) and daily access to an outdoor area (250 $m^2$). Each barn was equipped with at least one window to provide the animals with natural daylight inside the barn. The windows were installed outside the bird's visual field, so previous interactions with mirroring reflections were not likely. The birds were fed *ad libitum* (with "VoMiGo", Deuka, Deutsche Tiernahrung Cremer GmbH & Co KG, Düsseldorf, Germany) and had permanent access to water. All roosters were housed in single-sex groups with visual contact with females and had physical contact with hens for their first ten weeks of life. At the beginning of the experiments, all roosters were sexually mature and at least 19 weeks old. Individuals were marked with colored and numbered leg bands.

In a second bout, we repeated the mirror-audience test and added the mark test to the testing procedure, this time with 18 adult roosters of three different breeds: 9 Lohmann Brown (LB), 5 Bielefelder (B) and 4 Malines (M). The animals hatched and were raised at the Campus Frankenforst of the Agricultural Faculty of the University of Bonn (Königswinter, Germany), in a mixed-sex group until their 10th week of life. Afterwards, roosters were separated into a male-only-group under conditions of conventional free range. The barn had a size of 16 $m^2$ with daily access to an outdoor area of 106 $m^2$. Two windows (1.2 x 1.35 m and 0.8 x 1.15 m) allowed animals access to natural daylight inside the barn. The windows were out of the bird's visual field, so previous interactions with mirroring reflections were not likely. They had *ad libitum* access to food ("Allmash-L", Deuka, Deutsche Tiernahrung Cremer GmbH & Co KG,

Düsseldorf, Germany) and water. At the beginning of the experiments, the roosters were 30 weeks old and marked individually with colored leg bands.

All animals were completely naive to any experiments and mirrors.

## Experimental set-up

Two experiments on mirror self-recognition in male chickens were performed, the mirror-audience test and the mark test. The mirror-audience test was carried out twice in similar conditions, with different breeds and slightly different locations. The mark test involved a subset of the individuals of the second mirror-audience test.

Before the experiments, animals were gently caught in their home pen and transported in wooden boxes to the experimental arena in a separate room to reduce external stimuli. The animals were brought back to the barn immediately after their sessions were finished (maximum duration of separation from social group 45 min.). The experimental room had blank walls, no windows, and was evenly illuminated by a non-flickering light source (*first study*: daylight fluorescent lamp with UV-array and electronic control gear; *second study*: Spectra-lux®Plus, NL-T8 58W/840/G13, Radium, Germany). The arena (*first study*: 2 x 1.2 x 0.8 m, *second study*: 2 x 1 x 0.75 m, LxWxH) was built up from waterproof plywood boards with a floor of grey linoleum (*first study*) or green polyvinyl chloride floor (*second study*), respectively. It was located in the middle of the separate room, central under the light source, and subdivided into two, adjacent compartments (*first study*: 1 x 1.2 m; *second study*: 1 x 1 m). The two compartments could be separated from each other by a black, non-reflective wire mesh (1 x 3 cm mesh size), a wooden plate (1 x 0.8 m) or a mirror (1 x 0.8 m), depending on the experimental condition. The arena was covered by a net (8 cm mesh size) to prevent animals from leaving the arena.

Experimental and habituation sessions were recorded by a video camera, installed centrally above the arena on the ceiling (mirror-audience test) or attached to the edge of the arena walls (mark test). Video-recording software (*first study*: Viewer v3.0.1.241, Biobserve GmbH, Bonn Germany; *second study*: Debut Professional, v5.19, NCH Software, Canberra, Australia) recorded the sessions. Additionally, a microphone (*first study*: Yukon DSAS, Yukon Advanced Optics Worldwide, Vilnius, Lithuania; *second study*: EM 9600, the t.bone, Burgebrach, Germany) recorded the acoustic expressions of the animals during the mirror-audience test, which was further analyzed with the software Raven Pro (Cornell Lab of Ornithology, Ithaca, NY). For mirror-audience tests, a video projector projected a white area to the ceiling (*first study*: ceiling height 2.40 m, projection area 1.30 x 1.60 m) or an inserted ceiling (*second study*: white plate of 1 x 1.4 m, which hung 28 cm below the ceiling; ceiling height 2.73 m, presentation at 2.45 m, projection area 1 x 1.40 m), respectively. The inserted ceiling prevented the ceiling light from extending into the projection area and disturbing the perception of the projection. The ceiling light was left on during the experiment to prevent the roosters from being inactive due to darkness. Via PowerPoint (Microsoft® Office 2019, Redmond, WA, US), a hawk's silhouette flying diagonally from left to right and vice versa for 15 s (5 flights) could be projected towards the inserted ceiling (*first study*: 2.40 m distance between silhouette and floor, 35 cm hawk silhouette; *second study*: 2.45 m distance between silhouette and floor, 35 cm hawk silhouette). The hawk-silhouette was provisioned by the laboratory of Evans and Evans and used in previous studies [38].

## Procedure

**Habituation.**   Prior to testing, all animals underwent habituation with one habituation session per animal and day. In the first study, the animals were placed three times, for 15

minutes each, inside the empty arena with a white projection of the video projector onto the ceiling. In total, the animals in the first study received 45 minutes of mirror exposure before the beginning of the mirror-audience test. The 18 animals of the second bout passed through six habituation sessions to include habituation for the subsequent mark test. Here, habituation sessions also took 15 minutes each and included habituation in the empty arena (session 1), to the mirror (sessions 2–4 and 6) and a white projection on the ceiling without a mirror (session 5). Thus, animals in the second study also received 45 minutes (habituation sessions 2–4) of mirror exposure prior to the mirror-audience test and then another 15 minutes (habituation session 6) between the mirror-audience test and mark test. During habituation session 2 involving the mirror, four video files were not at their full-length (15 minutes) due to damage. Similarly, five video files in the sixth habituation session were damaged and incomplete. These videos were, therefore, excluded from statistical analysis.

Analog to the study of Parishar et al. [18], several behaviors were scored during habituation sessions, including (1) social responses: time in s spent in front of the mirror (head facing the mirror at 180˚ angle, see Parishar et al. [18]), number of aggressive or exploratory pecks towards the mirror or grid, fights towards the mirror, number of exploratory pecks in the environment e.g., wall or floor, number of times crowing, number of times of plumage ruffling, (2) contingency testing, which comprises behaviors performed in front of the mirror that include movements of the body to generate a visual-kinesthetic match between the individual's movement and its reflection in the mirror. Due to the lateral placement of the birds' eyes, head turns from side to side or from the midline in front to the side are used to explore the environment. Thus, the number of head turns in front of the mirror as well as head shaking was included as contingency testing. (3) Mark-directed and preening behavior was analyzed, including the number of times birds preened the mark region or other body regions while facing the mirror and the number of times birds preened the mark region or other bod regions while facing away from the mirror and (4) search responses: number of times birds turned clockwise or anticlockwise in front of the mirror.

**Experiment 1: Mirror-audience test.** The mirror-audience test was performed to investigate how roosters perceive their mirror image by exploiting their instinctive behavior of alarm calling by the combined presence of a predator and a conspecific. The day after the 5[th] habituation session, the mirror-audience test started to evaluate the audience-effect of alarm calling under four different conditions (Table 1): (A) the second compartment is empty, (B) a mirror is placed between compartments, (C) another rooster of the same breed is inside the second compartment, (D) a mirror is placed between compartments and another rooster of the same breed is placed behind the mirror in the second compartment. Condition B (mirror) was the main interest. The other conditions served as comparison and control conditions. For a subset

**Table 1. Schematic representation of the set-up of the four different conditions in the mirror-audience test.**

| Test compartment | Separation | | Back compartment |
|---|---|---|---|
| Test animal | (Acrylic glass) | Fence wire | **A**: Empty |
| | | | **B**: Mirror |
| | | | **C**: Rooster of the same breed |
| | | | **D**: Mirror + rooster of the same breed |

In Condition A the test animal was presented with an empty second compartment. In condition B, a mirror was placed between the two compartments with the reflecting surface toward the test animal. In condition C another rooster of the same breed as the test animal was placed in the second compartment, separated with fence wire from the compartment with the test animal. In condition D a mirror was placed between the two compartments with the reflecting surface toward the test animal and another rooster was placed in the second compartment behind the mirror to control for olfactory and auditory cues.

of the first study and the whole second study, the compartment-partition consisted of acrylic glass and wire mesh to minimize perceptual differences between the conditions with or without a mirror in all conditions. The last condition (D) was designed to control for a possible influence of olfactory and acoustical cues from the conspecific.

The mirror-audience test was performed in three consecutive 15-minute-sessions per day and animal; each rooster was in the arena once per day, either as a test or as a stimulus animal. The order of conditions for each rooster and the test-stimulus-pairs were assigned pseudo-randomly to exclude an effect of the order. Before placing the test animal inside the test compartment in conditions C and D, a stimulus animal of the same breed was placed inside the back compartment. An opaque, but light-transmissive acrylic glass (1 x 1 m) covered this compartment to prevent the stimulus animal from seeing the projection and thus being animated to elicit alarm calls, too. With the animal(s) inside the arena, a white screen was projected onto the inserted ceiling for 15 minutes to ensure that occurring behaviors during the following presentation could be ascribed to the predator presentation. After 15-minutes of white-screen presentation, a hawk's silhouette (projected wingspan 35 cm, flight speed 0.7 m/s) was projected flying diagonally for 15 seconds. This sequence of white-screen and predator presentations was repeated three times, resulting in three projection trials per animal and condition (alarm call, one-zero-sampling). The duration of 15 minutes between the predator presentation was chosen accordingly to experiments on alarm calls by Evans et al. [39]. All sessions were video recorded.

For the analyses of the alarm calls, all sounds elicited by the roosters during the experiment were recorded separately by a microphone and further analyzed with the sound analysis program "Raven". Sound data were analyzed and scored "blind" and independently by two observers with high inter-observer agreement (93.98%). When a rooster elicited an alarm call it was scored with "1", 'no alarm call' or other sounds not assigned to an alarm call were scored with "0".

The experimental procedure of the first study was nearly the same, except for the difference that 40 roosters (20 RL, 10 BR and 10 BLC) had been tested only under conditions A, B and C and without the acrylic glass partition between the two compartments, and additional 20 roosters (10 RL and 10 LB$_b$) were tested only under conditions C and D with the acrylic glass partition between compartments. An analysis of condition C with and without acrylic glass between compartments showed that the acrylic glass, and thus potential reflections, did not influence the number of alarm calls (U = 260000, p = 0.354). For this reason, the conditions with and without acrylic glass were pooled for further analysis.

**Experiment 2: Mark test.** Following the mirror-audience sessions, a 6$^{th}$ 15-minute-habituation to the mirror was conducted with each rooster in preparation for the mark test, which started the day after the 6$^{th}$ habituation (Fig 1). The procedure of the mark test was conducted

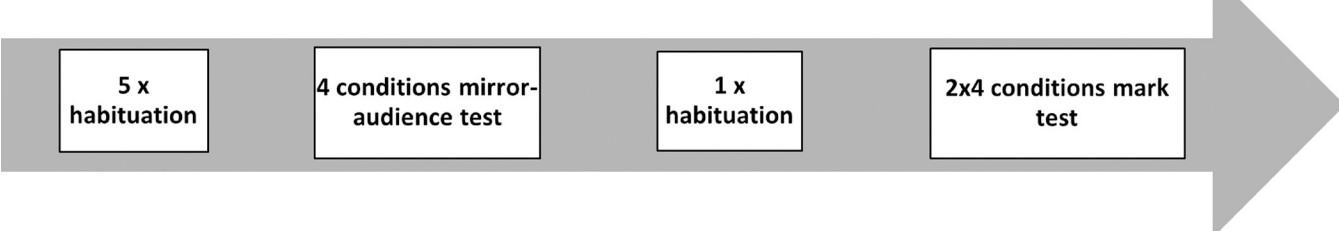

**Fig 1. Schematic representation of experimental procedure.** Shown is the sequence of habituation and test phases of an individual animal. See text for details e.g., on criteria and animal numbers.

according to the protocol by Prior et al. [7] and it took place in the same experimental room and arena as the mirror-audience test. The mark test consisted of four conditions: (1) mirror and colored mark, (2) mirror and sham mark, (3) no mirror and colored mark and (4) no mirror and sham mark. Each of the 18 roosters of the second study-data-collection ran each condition twice with the order of conditions per rooster assigned pseudo-randomly. It took eight days to test each animal once per day. Each session lasted 15 minutes. During this time the rooster was alone inside the arena in a separate testing room.

While Prior et al. [7] used stickers as markings, we used powder to minimize the possibility that the animals somehow feel disturbances on their feathers and because markings should be easily removable after experiments. To distinguish between different breeds, we opted for a pink powder (Gulal color powder, Pabo-GmbH, Mönchengladbach, Germany) since it offered the most contrast against the varying plumage colors. For sham marks, a transparent fixing powder (Eulenspiegel, Hadamar, Germany) was used. The use of such a sham mark ensured the avoidance of artifacts due to potential differences in handling between conditions. Markings were attached on the 'triangle-region' between the wattles just below the beak, which was out of the roosters' visible field (Fig 2). The handling procedure was the same in all four conditions: the test animal was taken to the experimental room where the colored or transparent mark, depending on the test condition, was attached to the bird's plumage. One experimenter gently held the animal with its neck in an upright position and put one hand flat below the beak to shield the rooster from seeing the fixing process. A second experimenter applied the colored or transparent powder on the described spot below the rooster's wattle. For easier application, the experimenter's finger has been moistened with water before picking up the powder, feathers have not been moistened. Immediately after the application of the mark or sham mark, the animal was placed inside the arena.

Depending on the test condition either a mirror or a non-reflective wooden plate was placed inside the arena. To avoid missing any behaviors towards the mirror, room lights were turned off before placing the animal inside or being picked out of the arena. Turning the lights on marked the beginning of a test session and the start of video recording, occurring immediately after the experimenter left the room.

For analysis, all observable behaviors during the sessions have been protocolled via video analysis, which resulted in a list of 14 behaviors: touching the mark or mark region with the beak, touching other parts of the body with the beak, scratching the head with the foot, pecking at the mirror, pecking at the floor, showing threatening gestures (e.g. setting up the neck feathers), crowing, fluffing up the plumage, shaking the head, flapping the wings, stretching, scratching the floor, resting, and sleeping. Behaviors not relevant to the question of this study were not further analyzed (e.g., resting or fluffing up plumage).

## Statistical analysis

The graphical representation of the results was done using SigmaPlot 14.0 (Systat Software Inc., Chicago, IL, USA). For statistical analyses, the program SPSS® Statistics 28 (IBM Corporation, Armonk, USA) was used. For all statistical tests, the significance-level α was set at $p < 0.05$. Significant differences are indicated in the figures as * for $p \leq 0.05$, ** for $p \leq 0.01$ and *** for $p \leq 0.001$. Data are presented with mean and standard deviation (M ± SD) for parametric tests and Median and interquartile range (Mdn and IQR) for non-parametric tests.

The comparison of behaviors between mirror and no-mirror sessions during habituation was calculated via non-parametric Mann-Whitney-U tests. Changes in the progression of behaviors during mirror-exposure sessions were analyzed using non-parametric Friedmann-tests.

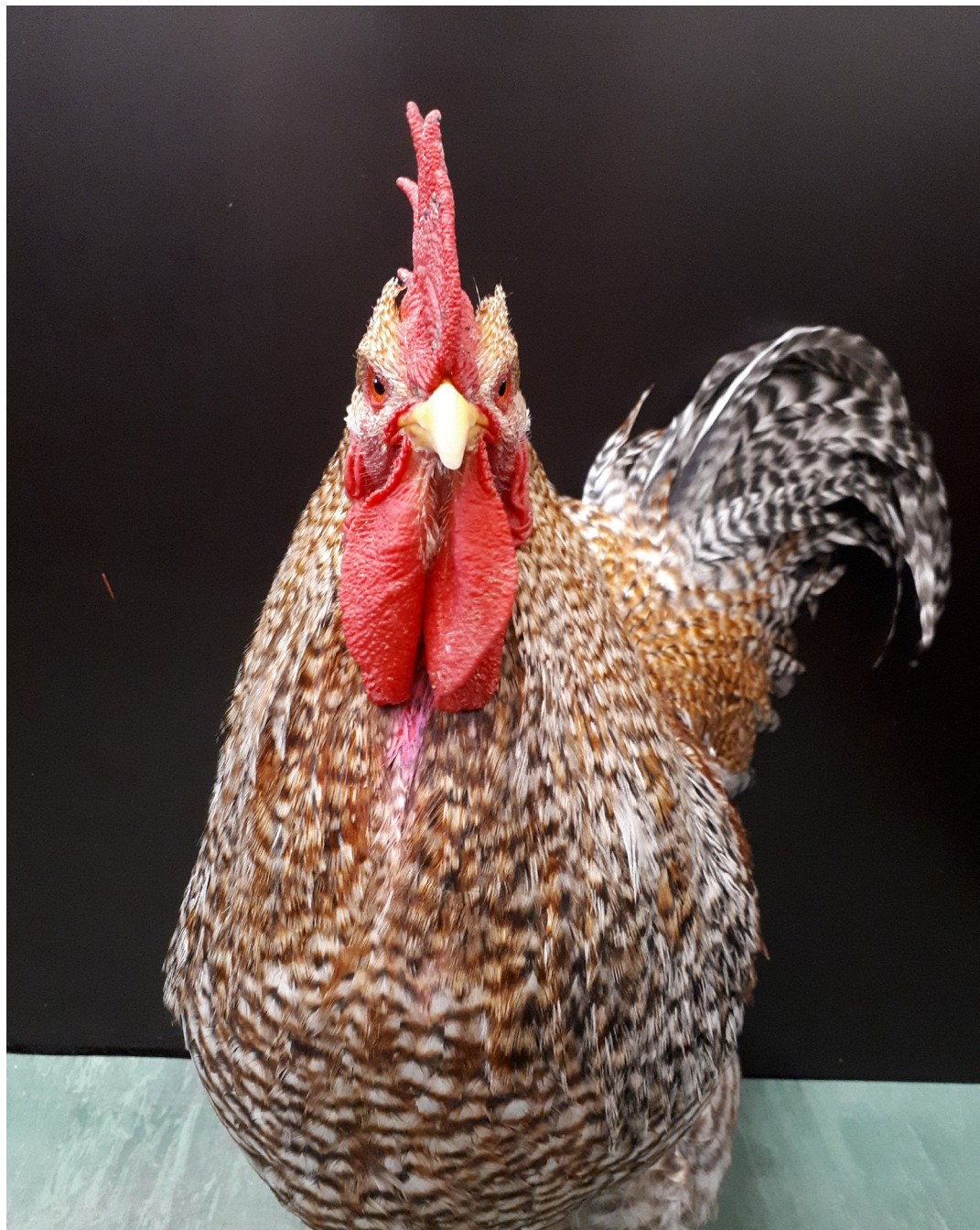

**Fig 2. Bielefelder rooster with mark.** The pink mark in the 'triangle-region' between the wattles was applied with colored starch powder.

For comparisons between the different conditions, in the mirror-audience test and the mark test a repeated measures ANOVA (rm-ANOVA) was used with conditions as dependent variables, followed by pairwise comparisons using Bonferroni-adjustment. Partial eta squared was used as a measure of effect size in the ANOVA. If sphericity was not given (p ≤ 0.05), Greenhouse-Geisser or Huynh-Feldt adjustments were applied, depending on ε (Greenhouse-

Geisser, if ε < 0.75; Huynh-Feldt, if ε > 0.75; following Girden [40]). For testing on the individual level in the mark test, a Chi-Square-test was chosen, giving Cramer's V.

### Ethics approval

The keeping of the animals complied with the order on the protection of animals and the keeping of production animals in Germany [41]. The Campus Frankenforst of the University of Bonn is approved as a trial farm (39600305-547/17) and the procedure of the experiments has been approved by the State Agency for Nature, Environment and Consumer Protection (LANUV; AZ 81–02.04.2019.A372).

## Results

### Habituation

When comparing frequencies of behaviors with and without the presence of a mirror during habituation, we found roosters exhibiting significantly more head turns in front of a mirror (Mdn = 25.00, IQR 15.00–45.00) than when the mirror was absent (Mdn = 16.00, IQR 11.25–24.00; n = 99, U = 1546.00, $p = 0.003$). Furthermore, pecks at the wall occurred more often in the presence of a mirror (Mdn = 1.00, IQR 0.00–6.00), than without a mirror (Mdn = 0.00, IQR 0.00–1.00; n = 99, U = 1581.00, $p < 0.001$). Roosters pecked more often at the grid, separating the potential stimulus and focus animal (Mdn = 0.00, IQR 0.00–2.75) than at the mirror (Mdn = 0.00, IQR 0.00–0.00; n = 99, U = 822.50, $p = 0.003$). All other behaviors (preening mark region in front of and away from mirror, preening other body regions in front of and away from mirror, time spent in front of mirror or grid, fights at mirror or grid, sleeping, peckings at floor, turning clockwise or anticlockwise, crowing, fluffing plumage and head shaking) occurred at similar incidences during mirror- and no-mirror-habituations (all $p > 0.05$, S1 Table). It is important to state that no bird was marked during habituation. The term "mark region" refers to the region between the wattles just below the beak, where the marks were painted in the subsequent mark-test.

As the decrease in social behaviors and the increase of contingency testing with increasing mirror experience is viewed as a prerequisite for MSR, the development of the above-mentioned behaviors was compared between the mirror habituation sessions. While the occurrence of the majority of behaviors did not change with the course over time during mirror exposure (S2 Table), there was a decrease in the number of head turns (n = 9, $\chi^2(3) = 8.42$, $p = 0.038$) from the 1st to the 4th mirror exposure ($p = 0.064$).

**Experiment 1: Mirror-audience test.** The mirror-audience test was conducted to test if roosters warn their mirror image with an alarm call or if they do not treat their reflection as a conspecific. This was evaluated under four conditions: (A) rooster + empty compartment, (B) rooster + mirror, (C) rooster + conspecific and (D) rooster + conspecific behind a mirror (Fig 3).

In total, we tested 68 roosters. Of these 68 roosters, 28 underwent all four conditions (A, B, C and D). Additional, 30 roosters were tested under conditions A, B and C and another 10 roosters underwent only conditions C and D. Based on one alarm call per predator presentation, each rooster could emit up to 3 alarm calls per condition resulting in a total number of 174 possible alarm calls in each of conditions A and B, 204 in C and 114 in D, due to the different numbers of tested animals per condition.

We compared the mean numbers of elicited alarm calls in conditions A, B, and C, testing 58 roosters, for the statistical analysis. This analysis revealed significantly different numbers of alarm calls between these three conditions (A: 0.29 ± 0.62; B: 0.43 ± 0.86; C: 1.33 ± 1.19; Huynh-Feldt F[1.691, 96.415] = 28.420, $p < 0.001$, $\eta_p^2 = 0.333$; Fig 4). Results clearly showed

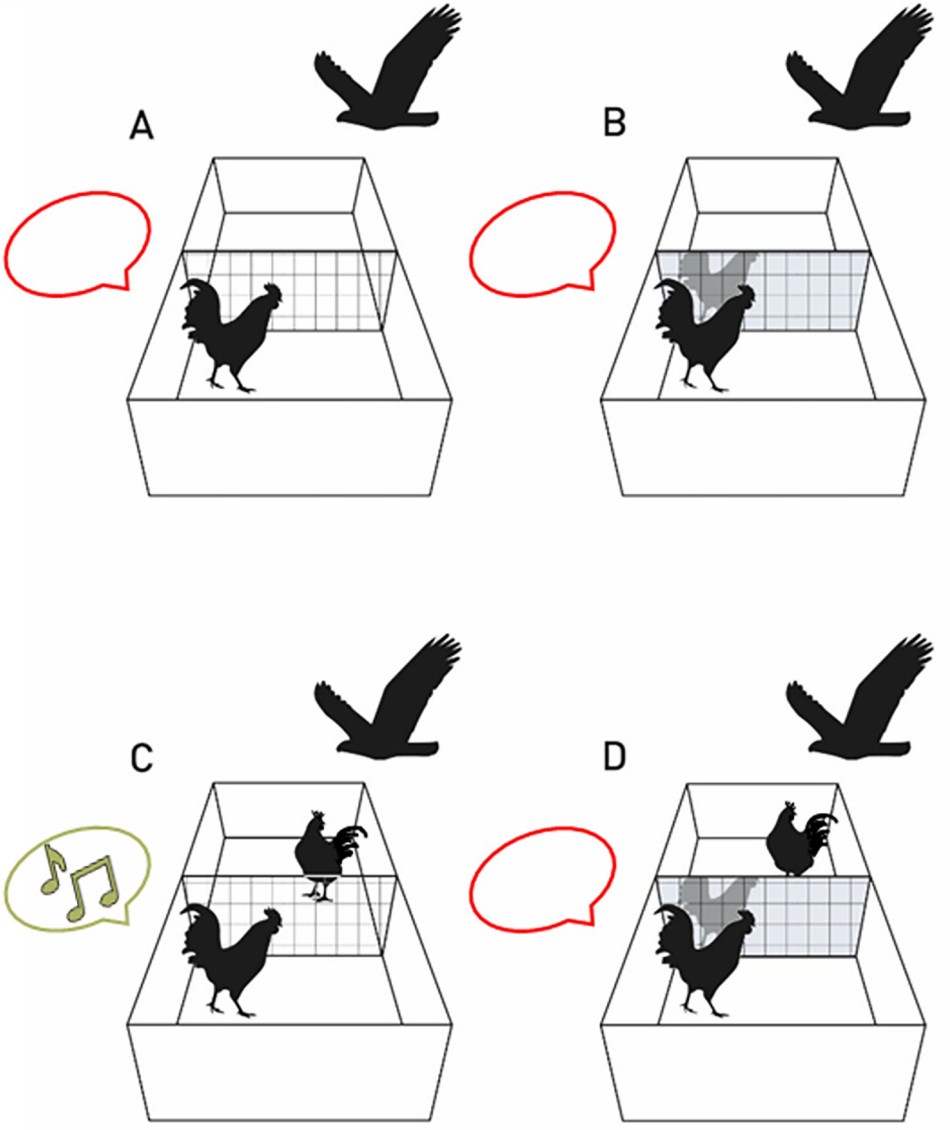

**Fig 3. Graphical representation of the four conditions of the mirror-audience test and its outcome.** The focus rooster stood in a longitudinal compartment with transparent acrylic glass (control for reflections) and wire mesh partition. Occasionally, the moving shadow of a passing bird of prey was projected onto the ceiling. The focus rooster did not emit alarm calls in conditions A (focus bird is alone), B (focus bird with its own mirror reflection), and D (focus bird with its own mirror reflection while another rooster is obstructed from view but located in the adjacent compartment). The alarm calls only occurred in condition C (focus bird with another rooster in the adjacent compartment).

that the presence of a conspecific (C) led to more frequent alarm calls than both the alone (A $p < 0.001$) and the mirror (B $p < 0.001$) condition. A comparison of the mirror condition (B) and the alone condition (A) revealed no significant results ($p = 1.00$).

As an additional control, we analyzed the subset of 28 individuals, tested under all four conditions, A, B, C, and D. This analysis clarified if, in condition D, olfactory or auditory cues of the conspecific behind the mirror had an effect on alarm calling. We found the same pattern of results as in the first analysis (Huynh-Feldt F[2.748, 74.186] = 6.442, $p = 0.001$, $\eta_p^2 = 0.193$; Fig 4). The presence of a conspecific behind the mirror (D: n = 28, 0.39 ± 0.74) did not animate

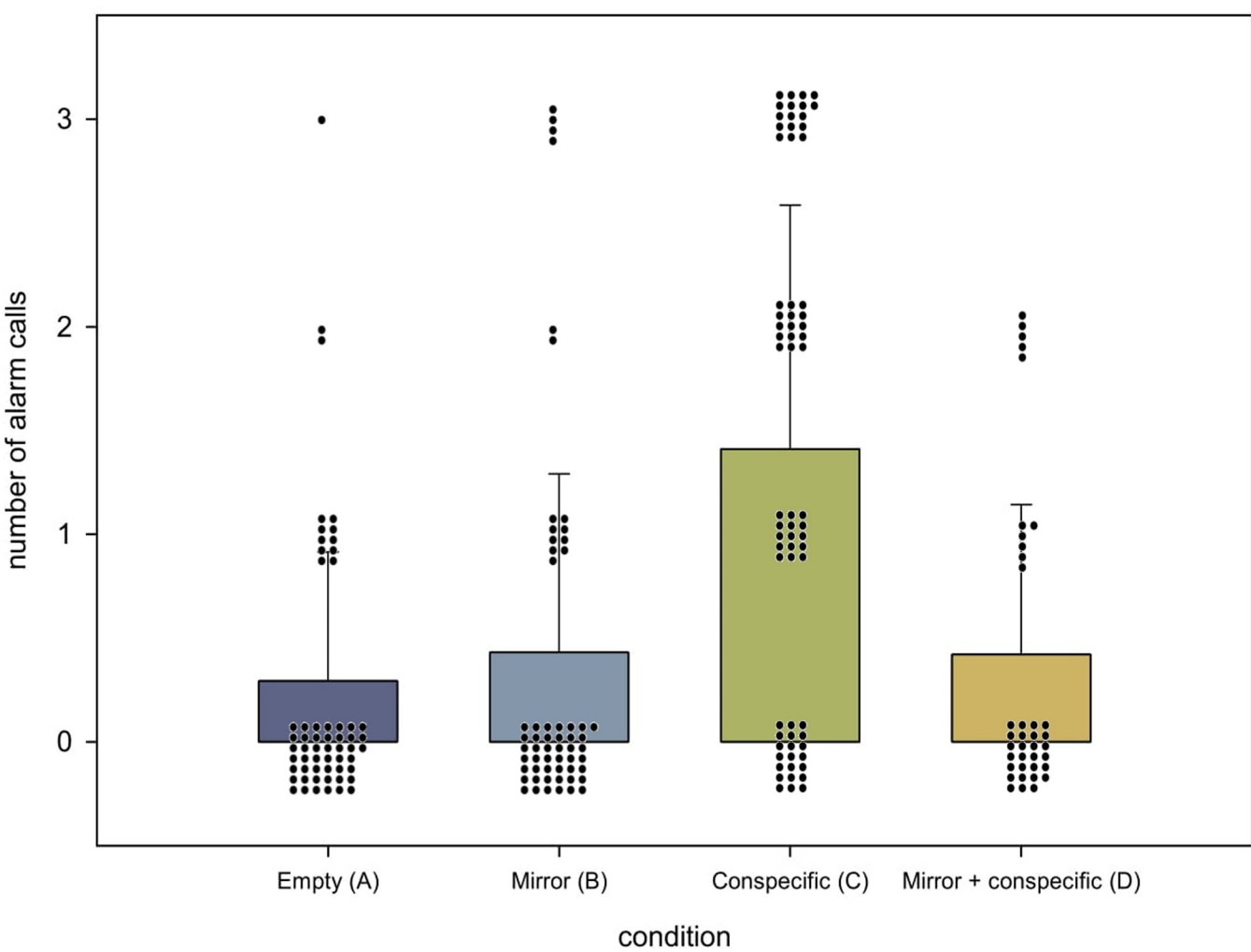

**Fig 4. Mean numbers ± SD of elicited alarm calls in the four different conditions.** Mean numbers ± SD of elicited alarm calls in each condition. Within each condition, the overlying dots represent every individual tested along with the number of alarm calls each individual had elicited in each condition (max. 3 alarm calls). Significant differences are indicated by stars, with *** for p < 0.001.

roosters to elicit alarm calls as often as in the presence of a visible conspecific (C: n = 28, 1.07 ± 1.09; *p* = 0.047). In contrast, the number of alarm calls in condition D (conspecific behind the mirror, n = 28, 0.39 ± 0.74) was nearly identical to those in condition A (empty compartment; n = 28, 0.25 ± 0.52; *p* = 1.00) and B (mirror: n = 28, 0.36 ± 0.73; *p* = 1.00).

## Experiment 2: Mark test

The mark test was conducted similarly to a previous study in magpies [7], and numbers of mark-directed as well as self-directed actions towards other parts of the body have been included in the analysis. The mark test was conducted with the second subset and applied to 18 birds of the previous study (9 LB, 4 M and 5 B). Behavior directed to the mark or sham-mark, as well as behavior directed to other parts of the body could be observed in all four conditions. The birds touched their body almost exclusively with their beak to groom their plumage and used their foot only to scratch their head in some cases. One bird never showed any mark-directed behavior (LB1). However, self-directed behavior to other parts of the body could be observed in all animals in at least one condition (Table 2).

**Table 2. Individual results of all conditions in the mirror mark test.**

| Subject | 1 –Mirror / Mark | 2 –Mirror / Sham mark | 3—No mirror / Mark | 4—No mirror / Sham mark |
|---|---|---|---|---|
| LB1 | 0 (0) | 0 (5) | 0 (2) | 0 (7) |
| LB2 | 0 (5)[a] | 0 (0) | 3 (2) | 0 (7) |
| LB3 | 1 (5) | 9 (10) | 5 (10) | 1 (7) |
| LB4 | 0 (0)[a] | 1 (2) | 1 (3) | 0 (0) |
| LB5 | 5 (10)[a] | 2 (14) | 1 (9) | 1 (6) |
| LB6 | 4 (8) | 1 (7) | 2 (4) | 0 (4) |
| LB7 | 10 (16) | 4 (16) | 2 (9) | 2 (7) |
| LB8 | 0 (0) | 1 (2) | 0 (7) | 2 (3) |
| LB9 | 3 (8) | 0 (3) | 1 (3) | 1 (2) |
| B1 | 2 (5) | 0 (5) | 3 (15) | 0 (3) |
| B2 | 3 (11) | 3 (17) | 3 (10)[a] | 4 (13) |
| B3 | 7 (14) | 13 (22) | 5 (14)[a] | 6 (12) |
| B4 | 8 (20) | 5 (15) | 3 (9) | 2 (5) |
| B5 | 0 (2) | 2 (6) | 8 (11) | 0 (2) |
| M1 | 4 (7) | 3 (7) | 0 (2) | 4 (12) |
| M2 | 0 (5) | 0 (22) | 2 (19) | 1 (7) |
| M3 | 3 (7) | 0 (2) | 0 (6) | 0 (3) |
| M4 | 7 (18) | 3 (6) | 5 (22) | 11 (24) |
| M ± SD | 3.47 ± 3.23 (8.40 ± 6.23) | 2.61 ± 3.48 (8.94 ± 7.01) | 2.25 ± 2.24 (8.31 ± 6.11) | 1.94 ± 2.84 (6.89 ± 5.61) |

Total number of behavioral actions towards the mark region (actions to other parts of the body are in brackets), presented as sum of trial periods 1 and 2. The last row presents mean (M) and standard deviation (SD) of mark-region-actions (and other-parts-of-the-body-actions) of all individuals.

[a] Indicates where the mark was accidentally misplaced or shifted in one of two trials due to the bird's prancing and thus was visible for the bird without the aid of a mirror, that is why these cases have been excluded from the analysis.

According to Prior et al. [7], we used the quantitative amount of mark-directed behaviors as a proportion of all behaviors directed to the body for analysis to correct for a bias of a general increase of self-directed actions. A comparison of the quantitative amounts of mark-directed behaviors revealed no significant differences between conditions (F$[3, 36]$ = 0.263, $p$ = 0.852, $\eta_p^2$ = 0.021; 1: 20.56% ± 15.16, 2: 17.43% ± 16.39, 3: 17.05% ± 14.45, 4: 15.81% ± 14.99), nor between breeds (F$[2, 64]$ = 0.528, $p$ = 0.592, $\eta_p^2$ = 0.016; LB: 18.24% ± 16.66, B: 21.29% ± 13.40, M: 16.92% ± 14.53).

Analysis on the individual level showed no significant correlation between touches of mark region or touches of other parts of the body and condition (all $p > 0.05$; S3 Table). In summary, there was no indication that roosters passed the mark test at group level or individually.

## Discussion

We departed from the idea that embedding the mirror self-recognition test into a species-specific ecological framework could potentially uncover hitherto unexpected cognitive abilities. To this end, we used roosters for our experiments for three reasons. First, these animals emit alarm calls when seeing aerial predators in the presence of other conspecifics [32, 33]. This species-specific behavior allows for a modified version of the MSR-test. Second, by showing the absence or presence of MSR in the classic and ecological conditions, we aimed to reveal that the contextual embedding of cognition can deliver very different result patterns for self-recognition. Third, chickens are possibly one of the least expected candidates for MSR. So, if roosters can differentiate between their own reflection and the sight of a conspecific, it is likely that this cognitive ability is much more widespread than previously assumed.

Our data shows that chickens emit significantly more alarm calls in front of a conspecific than in front of a mirror. Hereby, the number of calls emitted in the mirror condition was as low as when the rooster was alone, with no audience present. This cognitive differentiation seemed to depend on visual and not olfactory or auditory input since the animals with a conspecific behind the mirror also did not elicit more alarm calls. A similar finding has been shown by Dally et al. [42] who used the natural social behavior of re-caching of California (western) scrub jays to reveal behavioral differences between alone, social and mirror conditions. Scrub jays did not show the same behavioral responses towards their mirror image as with a living conspecific [42], similar to our chickens. On the contrary, they acted as when they were alone in the presence of their mirror image. This reaction shows these animals can discriminate between a living conspecific and their reflection in the mirror. However, the question arises if they perceive their mirror image as a "strange" conspecific that does not behave normally and mimics all movements, and thus, elicits a muted response than recognizing oneself in the mirror. Although the mirror-audience test offers another ecological approach, it is driven by the presentation of a predator that has to be identified urgently in nature, requiring rapid information processing in the brain [43]. This time pressure in information processing might also lead to a lacking or low reactivity towards the "stranger" in the mirror in the mirror condition.

Such an embedding of ecologically-minded testing methods for MSR or the discrimination of self from others was also an experimental aim in other studies like the "olfactory mirror" for grey wolves [44], dogs [45] or cichlid fish [46], the "chemical mirror" for garter snakes [47] and other studies addressing a species' ecologically relevant behavior [17, 19, 28, 31, 48]. These experiments reported that these species discriminate their own smell from that of others. As interesting this finding is, the interpretation of these results as a proof of self-recognition remains ambiguous [49].

Chicken use complex social cognitive strategies that contain elements of perspective-taking [36, 50]. They live within a close-knit community with complex, dynamic and stringent pecking orders, which require the ability to learn about one's own and others' positions within a hierarchy [36, 51]. Hogue et al. [51] demonstrated that hens do not only gain information about their own position within the hierarchy through their own social interactions, but they are also able to draw and assess information about their own ranking by visually observing interactions between other individuals by using transitive interference [52]. Chickens can also vary their signal structure within social context, e.g., in the context of tidbitting display: subordinate males manipulate the dual-component nature of tidbitting displays (vocal + behavior) by omitting the vocal component (avoiding attention from the dominant male) and then switching back to the vocal component when the dominant male is inattentive. This possibly requires the subordinate male to take the perspective of the dominant chicken to gain information about his attentional state [53]. In addition, males can use deception strategies to lure potential mates away from other males. For this, they emit food calls to attract females but do not provide food [54]. In response, females have developed counter-strategies as they avoid males that too often feigned the presence of food [55]. Thus, the ability of chickens to show MSR is less of a surprise when embedded within their social and ecological context.

Our chicken failed the classic mark-and-mirror test. But the mark test is only the final part of the procedures to probe the existence of MSR. Before it is applied, the development of self-related behaviors based on transitions from social behavior to contingency testing is usually observed in chimps in front of the mirror [3, 56]. We did not observe such a transition during mirror habituation sessions. Social behaviors like crowing, time spent in front of the mirror, fights towards the mirror, behavior associated with contingency testing like head shaking, and the preening of body parts occurred equally frequent during all habituation sessions,

irrespective of the presence of a mirror or the course over time. We only found significantly higher numbers of head turns in the presence of a mirror than when the mirror was absent. This finding could indicate an increased interest in exploring the mirror image [18]. Also, the exploratory pecks towards the arena walls were higher in the presence of a mirror, which may reflect increased interest in the environment although exploratory pecks at the mirror itself were less frequent than at the grid. Altogether, without the mirror-audience test none of our observations would have permitted the conclusion that roosters can differentiate their own mirror-image from conspecifics. Based on our results, we cannot completely rule out if our chickens regarded their mirror image as a strange conspecific that does not act "normal" and thus did not warn it. This question should be addressed in future research.

All tested roosters were naïve to mirrors and experimental procedures. This prerequisite addresses recent criticisms of the experimental procedures from other studies on MSR [57] and reflects the methodological criteria for MSR studies on gorillas developed by Murray et al. [58]. The number of study animals was 68 roosters for the ecological test and 18 for the classic mark test comparably high. In addition, pseudo-randomization of test conditions within experiments and between individuals excludes the effects of testing order. We used control conditions for olfactory and auditory cues in the ecological test and excluded to the best of our abilities tactile and olfactory cues in the mark test by using powder instead of sticker markings. As outlined in the method section, we also took all further precautions addressed as critical in the literature.

## Conclusions

Our findings imply that chickens clearly distinguish between their own reflection and the sighting of others. The classic MSR-procedure could not reveal this important difference. These data make it strikingly clear how much cognition is ecologically embedded. Consequently, when tested under appropriate contextual conditions, studies can uncover mental abilities indicative of self-reflection in the least expected places [59, 60]. Our results also point against a simple dichotomy of presence or absence of self-recognition and make it likely that this ability might not be as exclusive as suggested for half of a century.

## Supporting information

**S1 Table. Habituation analysis between mirror- and no-mirror sessions.** Results of the comparison of behaviors during habituation sessions with a mirror and sessions without a mirror. Given are Median (Mdn), interquartile-range (IQR), mean (M), standard deviation (SD) and test results of non-parametric Mann-Whitney-U-test.
(DOCX)

**S2 Table. Habituation analysis between multiple mirror sessions.** Results of the comparison of the different mirror exposure sessions 1–4 during habituation. Given are Median (Mdn), interquartile-range (IQR), mean (M), standard deviation (SD) and test results of non-parametric Friedmann-Test, followed by the Dunn-Bonferroni Post-hoc test giving z- and p-values.
(DOCX)

**S3 Table. Analyses at individual level for touches of the mark region and touches of other parts of the body during the mark test.**
(DOCX)

**S4 Table. Rawdata of all habituation sessions.** Frequency scoring of 14 behaviors and the duration of the time each animal spent facing the mirror. N/A (not applicable) for damaged

video files.
(XLSX)

**S5 Table. Rawdata of the mirror-audience test.** Number of elicited alarm calls for each rooster and test condition. N/A (not applicable) for animals that have not been tested in the specific condition.
(XLSX)

**S6 Table. Rawdata of the mark test.** Frequency scoring of 14 behaviors.
(XLSX)

## Acknowledgments

We would like to thank the animal caretakers and technicians of the Campus Frankenforst for the daily care and maintenance of the animals' health, as well as David Wilson, Department of Psychology, Memorial University of Newfoundland, St. John's, NL, Canada, for the provision of the predator silhouette as projection material. We also thank the students involved in the project for conducting the mirror-audience test in the first study with the first set of animals and Shannon Bräunig for proofreading the manuscript.

## Author Contributions

**Conceptualization:** Onur Güntürkün, Inga Tiemann.

**Data curation:** Sonja Hillemacher.

**Formal analysis:** Sonja Hillemacher, Sebastian Ocklenburg.

**Investigation:** Sonja Hillemacher, Inga Tiemann.

**Methodology:** Sonja Hillemacher, Sebastian Ocklenburg, Onur Güntürkün, Inga Tiemann.

**Resources:** Onur Güntürkün, Inga Tiemann.

**Supervision:** Onur Güntürkün, Inga Tiemann.

**Visualization:** Sonja Hillemacher.

**Writing – original draft:** Sonja Hillemacher, Onur Güntürkün, Inga Tiemann.

**Writing – review & editing:** Sonja Hillemacher, Sebastian Ocklenburg, Onur Güntürkün, Inga Tiemann.

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
