## [Decision Letter · Decision Letter 0]

21 Jun 2023

PONE-D-23-13584Roosters do not warn the bird in the mirror: The cognitive ecology of mirror self-recognitionPLOS ONE

Dear Dr. Hillemacher,

Thank you for submitting your manuscript to PLOS ONE. After careful consideration, we feel that it has merit but does not fully meet PLOS ONE’s publication criteria as it currently stands. Therefore, we invite you to submit a revised version of the manuscript that addresses the points raised during the review process. This is an interesting study that adds to the field of self recognition in animals. Although both reviewers provided very positive comments about the manuscript, they also raised issues regarding the background literature, interpretation and conclusions. The manuscript will also benefit from a thorough proof read to pick up on errors noted by both reviewers. It would also be useful to include the additional reference raised by reviewer 2, and the accompanying comments relating to the field and alternate interpretation of your results.

We look forward to receiving your revised manuscript.

Kind regards,

Thomas H. Burne

Academic Editor

PLOS ONE

Journal Requirements:

Reviewers' comments:

Reviewer's Responses to Questions

**Comments to the Author**

1. Is the manuscript technically sound, and do the data support the conclusions?

Reviewer #1: Yes

Reviewer #2: Yes

2. Has the statistical analysis been performed appropriately and rigorously? 

Reviewer #1: Yes

Reviewer #2: Yes

3. Have the authors made all data underlying the findings in their manuscript fully available?

Reviewer #1: Yes

Reviewer #2: Yes

4. Is the manuscript presented in an intelligible fashion and written in standard English?

Reviewer #1: Yes

Reviewer #2: Yes

5. Review Comments to the Author

Reviewer #1: This is an important and well-executed study that adds to our understanding of the phenomenon of mirror self-recognition. Currently, the literature on mirror self-recognition - its phylogenetic distribution and meaning - is extremely complex and becoming more so. This study helps to dismantle some of the issues surrounding the mixed results in the literature and is an example of an extremely clever way to probe mirror self-recognition using an ecologically valid framework. Well done. There are only a couple of minor points. First, the writing needs to be edited and checked for grammar and spelling. Second, the authors state that other studies, e.g., the "olfactory mirror" studies done with dogs and wolves, are valid measures of self-recognition. But they are not. They are simple discrimination tests and do not rise to the level of showing self-awareness. This statement should be modified accordingly.

Reviewer #2: This is a very nice study based on the natural behaviour of roosters to produce alarm calls when confronted with representations of aerial predators and utilising this knowledge to design an ecologically valid mirror test. It has been influenced by earlier studies on birds, but I was surprised that our own study using the re-caching behaviour of California (western) scrub-jays alone, when in private or in the presence of a mirror was not referenced, despite being the first to use a natural social behaviour as an index of behavioural differences between a social and mirror condition (Dally et al, 2010). I would therefore expect our study (details below) to be both referenced and discussed in the appropriate parts of the manuscript.

Dally JM, Emery NJ & Clayton NS (2010). Avian theory of mind and counter espionage by food-caching western scrub-jays (Aphelocoma californica). European Journal of Developmental Psychology, 7(1), 17-37.

Generally the current manuscript is clearly presented and well-written, but there are a few places where the text varies between current and past tense (lines 307, 317, 336) and there are a few spelling mistakes (line 317 'beginning', line 338 'mark test'), so I'd recommend that the authors take another pass over the manuscript to rectify these.

I have the same issue with the results presented in Experiment 1 as I did with our own study and hence why it's only published in the obscure European Journal of Developmental Psychology and a number of years after we first completed it. We don't know that the absence of a behavioural response in reaction to a mirror is the result of recognising that the bird in the mirror is the viewer (i.e. "that is me") or a response to a strange conspecific which doesn't move like other birds and which is always looking at the viewer, thereby eliciting a muted response because it is strange. I still don't know the answer to this and I don't think it has been addressed here either. I think it is wise (and the authors have continued this approach), to say that it opens up the potential for self-recognition in the species being studied, but it certainly doesn't provide categorical evidence that the species recognises itself in the mirror. Neither does the classic mark test either, despite this idea being dispelled by its original author.

I'm not sure I agree with the authors when they state they think it unlikely that the roosters' low reactivity to their mirror image is because they lack information in 2D about 3D stimuli, because roosters respond to the simple (silhouette) presentation of an aerial or ground predator because I would expect those two acts of perception to run on different cognitive systems. Antipredatory behaviour doesn't have the luxury of detailed computational assessment of a stimulus. Joseph LeDoux's work on the amygdala convincingly stressed two systems for assessing a threat work through the amygdala - an immediate assessment of danger allowing the animal to react quickly, even if the assessment is wrong - and a more calculated assessment that takes longer to process more complex information about the details of the stimulus, such as seeing a long, curvy shape moving in the grass, reacting as if it is a snake, but then see that it is actually a moving twine in the wind. Not reacting quickly if it was a predator could be the difference between life and death. As such, roosters are probably relying on the rapid threat assessment system to react to a silhouette of an aerial or ground predator quickly, so they can elicit an alarm call and make an escape.

Just a couple of specific questions:

It wasn't immediately clear how long each session was. I think 15min, but this could have been made a little clear. If so, does this mean that the roosters received a total of 1hr exposure to mirrors throughout the entire study (i.e. sessions 2, 3, 4 and 6)? This is quite brief. Even the initial studies in chimpanzees, they received many hours (I think 16, certainly overnight) of mirror exposure and their social responses gradually changed to self-directed responses over many hours. Do you think that you have provided the roosters with sufficient mirror exposure in order to see a difference? There was no real difference across mirror exposures in the limited time you provided, but this total may have been insufficient (especially when compared to other species). For own study, I believe we provided the scrub-jays with at least 14hr of mirror exposure before the study.

I wonder about the efficacy of the dye used in the mark test. Figure 3 suggests that the mark may have been difficult to see. Do you have any evidence, either in this study or unrelated studies using the dye, that the roosters could actually perceive the dye? Again, the problem with an absence of evidence study is that you don't know whether your subject animals could actually see what you wanted them to see. For example, did you perhaps dye some of their food and then train them to only peck at dyed food, then present them with dyed and non-dyed food to see if they only pecked at the dyed food?

I thought this was a great study and I would definitely like to see other studies like it.

6. PLOS authors have the option to publish the peer review history of their article (what does this mean?). If published, this will include your full peer review and any attached files.

Reviewer #1: No

Reviewer #2: **Yes: **Dr Nathan Emery

---

## [Author Response · Author response to Decision Letter 0]

9 Aug 2023

Responses to Editor and Reviewers:

Editor: Thank you for submitting your manuscript to PLOS ONE. After careful consideration, we feel that it has merit but does not fully meet PLOS ONE’s publication criteria as it currently stands. Therefore, we invite you to submit a revised version of the manuscript that addresses the points raised during the review process.

This is an interesting study that adds to the field of self-recognition in animals. Although both reviewers provided very positive comments about the manuscript, they also raised issues regarding the background literature, interpretation and conclusions. The manuscript will also benefit from a thorough proof read to pick up on errors noted by both reviewers. It would also be useful to include the additional reference raised by reviewer 2, and the accompanying comments relating to the field and alternate interpretation of your results.

Authors: Thank you for your constructive feedback to our manuscript. We fully understand the importance of addressing the issues raised by the reviewers to enhance the quality of our manuscript and to meet PLOS ONE's publication criteria. Furthermore, we conducted a thorough proofreading of the manuscript by a native speaker to correct any errors and improve the overall clarity and readability of our manuscript. We also added the rawdata files as supporting information.

Reviewer #1: This is an important and well-executed study that adds to our understanding of the phenomenon of mirror self-recognition. Currently, the literature on mirror self-recognition - its phylogenetic distribution and meaning - is extremely complex and becoming more so. This study helps to dismantle some of the issues surrounding the mixed results in the literature and is an example of an extremely clever way to probe mirror self-recognition using an ecologically valid framework. Well done. There are only a couple of minor points. First, the writing needs to be edited and checked for grammar and spelling. Second, the authors state that other studies, e.g., the "olfactory mirror" studies done with dogs and wolves, are valid measures of self-recognition. But they are not. They are simple discrimination tests and do not rise to the level of showing self-awareness. This statement should be modified accordingly.

Authors: Thank you for your valuable feedback to our manuscript. Our aim was to shed light on some of the issues surrounding the mixed results in the literature, and we appreciate that you recognize the importance of our study. Your comments on some weak points in our manuscript are very helpful. We carefully checked and revised the whole manuscript for grammar and spelling mistakes and made also a proofread by a native speaker.

Furthermore, we acknowledge your remark regarding the "olfactory mirror" studies conducted with dogs and wolves. We agree that the results of these studies did not necessarily provide explicit evidence for MSR. We modified out statement accordingly. 

Lines 516-522: “Such an embedding of ecologically-minded testing methods for MSR or the discrimination of self from others was also an experimental aim in other studies like the “olfactory mirror” for grey wolves [44], dogs [45] or cichlid fish [46], the “chemical mirror” e.g., for garter snakes [47] and other studies addressing a species’ ecologically relevant behavior [17,19,28,31,48]. These experiments reported that these species discriminate their own smell from that of others. As interesting this finding is, the interpretation of these results as a proof of self-recognition remains ambiguous [49].”

Reviewer #2: This is a very nice study based on the natural behaviour of roosters to produce alarm calls when confronted with representations of aerial predators and utilising this knowledge to design an ecologically valid mirror test. It has been influenced by earlier studies on birds, but I was surprised that our own study using the re-caching behaviour of California (western) scrub-jays alone, when in private or in the presence of a mirror was not referenced, despite being the first to use a natural social behaviour as an index of behavioural differences between a social and mirror condition (Dally et al, 2010). I would therefore expect our study (details below) to be both referenced and discussed in the appropriate parts of the manuscript.

Dally JM, Emery NJ & Clayton NS (2010). Avian theory of mind and counter espionage by food-caching western scrub-jays (Aphelocoma californica). European Journal of Developmental Psychology, 7(1), 17-37.

Authors: Dear Dr. Emery, we thank you for your insightful comments on our study. We greatly appreciate your positive feedback. We apologize for the oversight in not referencing and discussing your study on California (western) scrub-jays' re-caching behavior when in private or in the presence of a mirror and acknowledge the importance of your work.

We now made the necessary revisions to include proper referencing and discussion of your study in the appropriate parts of our manuscript. We thank you for bringing this to our attention and incorporated your study into our discussion and appropriately credit your contribution to the understanding of mirror-related behaviors in birds. 

Lines 494-507: “Our data shows that chickens emit significantly more alarm calls in front of a conspecific than in front of a mirror. Hereby, the number of calls emitted in the mirror condition was as low as when the rooster was alone, with no audience present. This cognitive differentiation seemed to depend on visual and not olfactory or auditory input since the animals with a conspecific behind the mirror also did not elicit more alarm calls. A similar finding has been shown by Dally et al. [42] who used the natural social behavior of re-caching of California (western) scrub jays to reveal behavioral differences between alone, social and mirror conditions. Scrub jays did not show the same behavioral responses towards their mirror image as with a living conspecific [42], similar to our chickens. On the contrary, they acted as when they were alone in the presence of their mirror image. This reaction shows these animals can discriminate between a living conspecific and their reflection in the mirror. However, the question arises if they perceive their mirror image as a “strange” conspecific that does not behave normally and mimics all movements, and thus, elicits a muted response than recognizing oneself in the mirror.”

Reviewer #2: Generally, the current manuscript is clearly presented and well-written, but there are a few places where the text varies between current and past tense (lines 307, 317, 336) and there are a few spelling mistakes (line 317 'beginning', line 338 'mark test'), so I'd recommend that the authors take another pass over the manuscript to rectify these.

Authors: We apologize for these oversights and assure you that we thoroughly addressed these concerns during the revision process. We carefully reviewed the entire manuscript to ensure consistency in the use of tense throughout and conducted a thorough spell-check together with a native speaker to correct any misspellings and ensure the accuracy of terminology.

Reviewer #2: I have the same issue with the results presented in Experiment 1 as I did with our own study and hence why it's only published in the obscure European Journal of Developmental Psychology and a number of years after we first completed it. We don't know that the absence of a behavioural response in reaction to a mirror is the result of recognising that the bird in the mirror is the viewer (i.e. "that is me") or a response to a strange conspecific which doesn't move like other birds and which is always looking at the viewer, thereby eliciting a muted response because it is strange. I still don't know the answer to this and I don't think it has been addressed here either. I think it is wise (and the authors have continued this approach), to say that it opens up the potential for self-recognition in the species being studied, but it certainly doesn't provide categorical evidence that the species recognises itself in the mirror. Neither does the classic mark test either, despite this idea being dispelled by its original author.

Authors: As you rightly pointed out, there could be various reasons for the observed lack of response to the mirror. It is indeed challenging to differentiate between self-recognition and the response to a strange conspecific that behaves “strange” and maintains constantly eye contact with the viewer. We acknowledge the limitations in our study and the potential ambiguity in interpreting the results. In line with your suggestion, we have maintained a cautious approach in our discussion by highlighting that the findings open up the potential for self-recognition in the species under study, rather than making definitive claims of self-recognition. Recognizing the complexities involved in studying self-recognition, we believe that our work contributes to the broader understanding of this phenomenon and provides a foundation for further investigations. We concur with your view that the classic mark test also has its limitations and may not be a definitive measure of self-recognition, despite earlier claims suggesting otherwise. The study of self-recognition is a complex and ongoing field, and we are aware of the need for further research and refinement in experimental methodologies to gain deeper insights into this cognitive ability.

Once again, we appreciate your valuable input and reflections on the matter. Your comments help strengthen the scientific discourse and encourage us to approach the topic of self-recognition with appropriate caution and openness to further exploration.

Lines 555-560: “Altogether, without the mirror-audience test, none of our observations would have permitted the conclusion that roosters can differentiate their own mirror-image from conspecifics. Based on our results, we cannot completely rule out if our chickens regarded their mirror image as a strange conspecific that does not act “normal” and thus did not warn it. This question should be addressed in future research.”

Reviewer #2: I'm not sure I agree with the authors when they state they think it unlikely that the roosters' low reactivity to their mirror image is because they lack information in 2D about 3D stimuli, because roosters respond to the simple (silhouette) presentation of an aerial or ground predator because I would expect those two acts of perception to run on different cognitive systems. Antipredatory behaviour doesn't have the luxury of detailed computational assessment of a stimulus. Joseph LeDoux's work on the amygdala convincingly stressed two systems for assessing a threat work through the amygdala - an immediate assessment of danger allowing the animal to react quickly, even if the assessment is wrong - and a more calculated assessment that takes longer to process more complex information about the details of the stimulus, such as seeing a long, curvy shape moving in the grass, reacting as if it is a snake, but then see that it is actually a moving twine in the wind. Not reacting quickly if it was a predator could be the difference between life and death. As such, roosters are probably relying on the rapid threat assessment system to react to a silhouette of an aerial or ground predator quickly, so they can elicit an alarm call and make an escape.

Authors: You raised a crucial point regarding the potential different cognitive systems involved in antipredatory behavior and the perception of stimuli in 2D or 3D. We agree that antipredatory behavior likely relies on rapid threat assessment systems, allowing animals to react quickly to potential dangers, even if the assessment is not detailed. Considering this distinction, we recognize that the roosters' responses to simple predator silhouettes and their lack of reactivity to their mirror image may indeed be governed by different cognitive processes. While antipredatory behavior may not require detailed computational assessment, self-recognition and responses to mirror images may involve more complex cognitive mechanisms. In light of your comments, we acknowledge the need for a more nuanced interpretation of the roosters' behavior in our study. It is plausible that the roosters' low reactivity to their mirror image may be influenced by the rapid threat assessment system, whereas self-recognition and responses to mirrors could require additional cognitive processing. We appreciate your insights and will consider these points in our discussion of the results. 

Lines 508-512: “Although the audience test offers another ecological approach, it is driven by the presentation of a predator that has to be identified urgently in nature, requiring rapid information processing in the brain [43]. This time pressure in information processing might also lead to a lacking or low reactivity towards the “stranger” in the mirror condition.” 

Reviewer #2: Just a couple of specific questions: 

It wasn't immediately clear how long each session was. I think 15min, but this could have been made a little clear. If so, does this mean that the roosters received a total of 1hr exposure to mirrors throughout the entire study (i.e. sessions 2, 3, 4 and 6)? This is quite brief. Even the initial studies in chimpanzees, they received many hours (I think 16, certainly overnight) of mirror exposure and their social responses gradually changed to self-directed responses over many hours. Do you think that you have provided the roosters with sufficient mirror exposure in order to see a difference? There was no real difference across mirror exposures in the limited time you provided, but this total may have been insufficient (especially when compared to other species). For own study, I believe we provided the scrub-jays with at least 14hr of mirror exposure before the study.

Authors: We apologize for any confusion in not clearly indicating the session duration in the manuscript. Each habituation session in our study lasted for 15 minutes, and you are correct in noting that the roosters received a total of 1 hour of exposure to mirrors for habituation (sessions 2, 3, 4, and 6) prior to the mark test. We tried to make this clearer in the specific part in the manuscript. That said, it is important to track the animal’s behavior across habituation sessions. Maybe due to the domestication status of chickens, here, the roosters were calm after three to four sessions. 

Lines 205-215: “Prior to testing, all animals underwent habituation with one habituation session per animal and day. In the first study, the animals were placed three times, for 15 minutes each, inside the empty arena with a white projection of the video projector onto the ceiling. In total, the animals in the first study received 45 minutes of mirror exposure before the beginning of the mirror-audience test. The 18 animals of the second bout passed through six habituation sessions to include habituation for the subsequent mark test. Here, habituation sessions also took 15 minutes each and included habituation in the empty arena (session 1), to the mirror (sessions 2-4 and 6) and a white projection on the ceiling without a mirror (session 5). Thus, animals in the second study also received 45 minutes (habituation sessions 2-4) of mirror exposure prior to the mirror-audience test and then another 15 minutes (habituation session 6) between the mirror-audience test and mark test.”

Reviewer #2: I wonder about the efficacy of the dye used in the mark test. Figure 3 suggests that the mark may have been difficult to see. Do you have any evidence, either in this study or unrelated studies using the dye, that the roosters could actually perceive the dye? Again, the problem with an absence of evidence study is that you don't know whether your subject animals could actually see what you wanted them to see. For example, did you perhaps dye some of their food and then train them to only peck at dyed food, then present them with dyed and non-dyed food to see if they only pecked at the dyed food?

I thought this was a great study and I would definitely like to see other studies like it.

Authors: Again, thank you for raising this issue. Regarding the efficacy of the dye used in the mark test, we agree that the visibility of the mark for the experimental subjects is crucial to obtain reliable results. We can only provide evidence from individual observations regarding the roosters' perception of the dye. For example, rooster LB2, for whom the mark was set too low and thus within his visual field in one of the two mirror+mark conditions and who preened the mark region significantly more often in this condition than in the other conditions (with correctly placed mark). Due to the misplacement of the mark, this condition for this specific rooster was obviously excluded from the analysis and is therefore not included in the numbers in Table 2. However, it provides a clue that the mark mattered when directly seen. In the repetition of the mirror+mark condition for LB2 with this time correctly placed mark (see table 2), LB2 did not preen the mark region at any time, while in the condition with the misplaced mark LB2 preened the mark region 5 times. The first of five times preening the mark region occurred when LB2 was facing away from the mirror, which may indicate that the rooster could indeed see the mark easily without the aid of a mirror.

---

## [Editor Report · Decision Letter 1]

30 Aug 2023

Roosters do not warn the bird in the mirror: The cognitive ecology of mirror self-recognition

PONE-D-23-13584R1

Dear Dr. Hillemacher,

We’re pleased to inform you that your manuscript has been judged scientifically suitable for publication and will be formally accepted for publication once it meets all outstanding technical requirements.

Kind regards,

Thomas H. Burne

Academic Editor

PLOS ONE

Additional Editor Comments (optional):

Many thanks for taking the time to address all of the reviewer comments.
---

## [Editor Report · Acceptance letter]

26 Sep 2023

PONE-D-23-13584R1 

Roosters do not warn the bird in the mirror:
The cognitive ecology of mirror self-recognition 

Dear Dr. Hillemacher:

I'm pleased to inform you that your manuscript has been deemed suitable for publication in PLOS ONE. Congratulations! Your manuscript is now with our production department. 

Kind regards, 

on behalf of

Professor Thomas H. Burne 

Academic Editor

PLOS ONE